# Providers' experiences with abortion care: A scoping review

**B. Dempsey**[1], **S. Callaghan**[1,2], **M. F. Higgins**[1,2]*

**1** UCD Perinatal Research Centre, National Maternity Hospital, University College Dublin, Dublin, Ireland,
**2** National Maternity Hospital, Dublin, Ireland

\* mary.higgins@ucd.ie

## Abstract

### Background

Induced abortion is one of the most common gynecological procedures in the world, with as many as three in every ten pregnancies ending in abortion. It, however, remains controversial. The objective of this scoping review was to explore and map existing literature on the experiences of those who provide abortion care.

### Methods and findings

This exploratory review followed the Levac et al. guidelines and was reported in accordance with the PRISMA-ScR checklist. CINAHL, Cochrane, EMBASE, PsycInfo, PubMed, and Web of Science were used to identify peer-reviewed, original research articles published on providers' experience of abortion. We identified 106 relevant studies, which include a total sample of 4,250 providers from 28 countries and six continents. Most of the studies were qualitative ($n = 83$), though quantitative ($n = 15$) and mixed methods ($n = 8$) studies were also included. We identified two overarching themes: (1) Providers' experiences with abortion stigma and (2) Providers' reflections on their abortion work. Our findings suggest that providers from around the world experience challenges within society and their communities and workplaces which reinforce the stigmatization and marginalization of abortion and pose questions about the morality of this work. Most, however, are proud of their work, believe abortion care to be socially important and necessary, and remain committed to the provision of care.

### Conclusions

The findings of this review provide a comprehensive overview on the known experiences of providing abortion care. It is a key point of reference for international providers, researchers, and advocates to further this area of research or discussion in their own territories. The findings of this review will inform future work on how to support providers against stigmatization and will offer providers the chance to reflect on their own experiences.

**Data Availability Statement:** All relevant data are within the paper and its Supporting information file.

**Funding:** This work was supported by the National Maternity Hospital, Dublin, Ireland in fulfillment of an academic qualification to be completed by the

lead author. The funders had no role in study design, data collection and analysis, decision to publish, or preparation of the manuscript.

**Competing interests:** We declare no financial conflicts of interest regarding this piece of work, though acknowledge that one author (Dr. Mary F. Higgins) campaigned for repeal during the 2018 referendum and is a practicing abortion provider in the Republic of Ireland. Additionally, two authors (BD and MFH) are involved in other research studies exploring providers' experiences with abortion care in the Republic of Ireland. This does not alter our adherence to PLOS ONE policies on sharing data and materials.

## Introduction

Induced abortion is one of the most common gynecological procedures in the world. Between 2015 and 2019, an estimated 73.3 million global induced abortions of unintended pregnancies occurred per year [1]. This equates to approximately three in every ten pregnancies ending in abortion and an average of 39 annual abortions per 1,000 women of reproductive age [1]. Despite the common nature of abortion, it remains controversial, and many studies have explored the various barriers to accessible care and the negative attitudes on abortion care. Abortion stigma is a prominent example in the literature. Originally defined as *"a negative attribute ascribed to women who seek to terminate a pregnancy that marks them, internally or externally, as inferior to ideals of womanhood"* [2], abortion stigma was later expanded to also explore how advocates and service providers may be negatively impacted by the *"shared understanding that abortion is morally wrong and/or socially unacceptable"* [3, 4]. It has been suggested that abortion is stigmatized as it violates the *"feminine ideals"* of motherhood, results in the *"murder"* of a fetus, is unfairly targeted by restrictive legislation, is seen as dirty or unhealthy, and that stigma is a prominent tool proliferated by anti-abortion groups [4].

Seminal studies in this area have explored the experiences of those who provide abortion care. In 1978, Carole Joffe first described abortion care as *"dirty work"* [5, 6]. Dirty work, first proposed by Everett Hughes in 1951 [7], denotes professions that are socially, physically, and/ or morally tainted. Such professions may be described as *"distasteful, disgusting, dangerous, demeaning, immoral, or contemptible"*, while also being viewed as necessary by society [8, 9]. By association, employees in such professions may become seen as *"dirty workers"* and may experience stigmatization related to their work. In the case of abortion, social taint may refer to the providers' interaction with the stigmatized individuals accessing care, physical taint may relate to the handling of fetal remains, and moral taint may relate to the ongoing debate regarding the unborn's right to life. Of these, moral taint is believed to be the most damaging [10]. We acknowledge that referring to a profession as "dirty" is inherently stigmatizing, and we will hereafter refer to abortion care as "stigmatized work"

Despite these challenges, it has been noted that staff employed in stigmatized professions generally hold positive social identities around their work [8, 10]. Ashforth and Kreiner proposed a theoretical model explaining that workgroup cultures within these professions aide in developing these positive "work role identities", as a strong workgroup culture promotes the use of social weighting techniques and occupational ideologies [8, 10]. Social weighting techniques discredit those who share negative thoughts about their work, while supporting those who share positive thoughts. Additionally, those in such work may selectively compare themselves to others who may be more stigmatized. Occupational ideologies include reframing, recalibrating, and refocusing. Reframing is the action of transforming the meaning of the profession from stigmatized to honorable work that positively adds to society. Recalibrating seeks to augment the more socially acceptable aspects of the profession and reduce the importance of less acceptable or stigmatized aspects. Refocusing is the action of magnifying the recalibrated features of the work. Ashforth and Kreiner argue that there is a reciprocal relationship between social weighting techniques and occupational ideologies, as both help to *"externalize or attribute the dirty work stigma to the ignorance or malevolence of outsiders"* [8]. Ultimately, these strategies help staff to feel as though they are *"good people doing good work"* [8].

No review to date has sought to scope and map the existing evidence on the lived experiences of those who provide abortion care. The aim of this review is to address this gap, paying close attention to how the providers' experiences relate to Ashforth and Kreiner's model of stigmatized work [8, 10].

## Materials and methods

In designing this review, various methodologies were considered. Given the broad nature of the research aim, a scoping review was considered most appropriate [11, 12]. This allowed for an open exploration of the available literature to curate a narrative review on what is known about the experience of providing abortion care. Additionally, given our broad aim to explore providers' experiences, we were not sure exactly what would be uncovered in the review, and we did not specify outcomes of interest. We believe that the findings of this scoping review may lay the groundwork for future systematic reviews which seek to explore specific experiences that the providers have in more detail. To maintain rigor and replicability in this review, we followed guidance from the *Preferred Reporting Items for Systematic reviews and Meta Analyses extension for Scoping Review* (PRISMA-ScR) tool [13] (S1 Table). Further, we followed the Levac *et al.* guidelines [14, 15], which include six stages: (1) identifying the research question, (2) identifying relevant studies, (3) study selection, (4) charting the data, (5) collating, summarizing, and charting the results, and (6) consultation. Each stage is described briefly; refer to our open-access protocol for more information [16].

### Stage 1—Identifying the research question

We devised the research question to guide the review by looking to relevant studies and reflecting on their findings [14, 17]. We phrased the question to explore the *experience of providing abortion care*, purposefully choosing the ambiguous term *experience* to illicit data on both positive and negative experiences related to abortion work. Additionally, we made the decision to use the broad term *providing abortion care* to include any individual who is directly involved, clinically or non-clinically, in the care of patients accessing care. This led to the following research question:

- What is the lived experience of individuals who are directly involved in the provision of comprehensive abortion care?

### Stage 2—Identifying relevant articles

In stage two, we decided on the eligibility criteria, databases, and search strategy for the review.

**Eligibility criteria.** In line with the research question, we searched for citations which focused on the experiences of those who have direct contact with people who access comprehensive abortion care services. We excluded studies that examined patients' experience of abortion, technical aspects of abortion care, or providers' experience of post-abortion care. Due to study constraints, we only included original research articles published in peer-reviewed journals in the English language, excluding all citations that did not meet this. We did not set restrictions for the year, country, or indication for abortion. From inception, we did not plan to conduct a detailed cross-cultural or temporal analysis of the providers' experiences due to time constraints. For this reason, we chose to include studies regardless of year of publication, as we were interested in exploring providers' experiences with the services at that time, rather than exploring how changes to the laws may affect their experiences.

**Databases.** To identify articles, we conducted a systematic search in six electronic databases: CINAHL, the Cochrane Library, Embase, PsycInfo, PubMed, and Web of Science.

**Search strategy.** We designed the search strategy for the electronic databases using the PCC framework (Population, Concept, and Context), as recommended by the Joanna Briggs Institute [12] (Table 1). In addition to searching for articles in the electronic databases, we

**Table 1. PCC elements for the study selection criteria, including an added *"exclusion"* Boolean operator.**

| Participant | provider* OR "healthcare professional*" OR "health professional*" OR "healthcare worker*" OR "health worker*" OR Clinician* OR midwi* OR nurse* OR obstetric* OR gynaecolog* OR gynecolog* OR OBGYN OR physician* OR doctor* OR practitioner* |
|---|---|
| **Concept** | experienc* OR stigma* OR discrimin* OR prejudic* OR violenc* |
| **Context** | abortion* OR "termination of pregnan*" |
| ***Exclusion**** | "spontaneous abortion" OR miscarriage* OR ectopic |

* Though not included in the PCC framework, the "Exclusion" operator was included when piloting the search strategy to reduce the large number of irrelevant papers.

searched for unidentified articles in the reference lists of studies that were deemed eligible for inclusion and hand-searched relevant journals.

## Stage 3—Study selection

Stage 3 was to search for articles. After downloading the citations from our database search and removing duplicates, a title screen was conducted. All duplicates were removed by the lead author (BD). For the title screen, BD met with one of the co-authors (SC or MFH) to review titles as a pair. The authors discussed each title and agreed to include or exclude each citation. Once title screening had concluded, BD then conducted an abstract review on the remaining citations. SC and MFH each independently reviewed 15% of all abstracts, meaning that 30% of the abstracts were independently reviewed by a second author. No discrepancies were found between the articles included during the first review by BD and second review by SC and MFH. Then, BD conducted an independent full-text review on the remaining citations. Again, SC and MFH independently reviewed 15% of the remaining citations, and no discrepancies were found among the 30% of articles that had been reviewed in full.

## Stage 4—Charting the data

Following guidance from the JBI [11], we created a table to extract information relevant to the review question (Table 2). Prior to beginning the full charting process, we conducted a preliminary analysis with a list of 32 articles that we included following the full-text review. This initial step was taken to simplify the analysis process given the large number of studies identified by the search process and the large amount of qualitative data to be extracted and processed as part of the review. Specifically, we followed guidance from Thomas and Harden on conducting a thematic synthesis on the findings of these initial studies [18]. We began by extracting findings related to providers' experiences with abortion care from each of these studies into an Excel file. As this was an exploratory review, we did not pre-determine what outcomes we were looking for. Rather, we reviewed each study for inclusion based on the broad criteria that it explored providers' experiences with care, and we charted all information that related to direct lived experiences that the providers shared. We then coded these charted passages line-by-line and collated the codes into descriptive themes. This led to the creation of 10 descriptive themes, each with sub-themes. We then started charting relevant passages from the remaining articles into the corresponding descriptive theme and sub-theme. During the full charting exercise, we reviewed, refined, and expanded on these descriptive themes and we ended the process with 13. Article reference details and information on the study context and design were also charted.

**Table 2. Charting elements that were devised by the research team to guide the data charting process.**

| Charting elements | Characteristics of the study |
|---|---|
| Reference details | Article reference number (to be given to each article by research team) |
| | Study title |
| | Author(s) |
| | Year |
| | Journal |
| | DOI (Digital Object Identifier) |
| Study Context | Study Aim(s)/Objective(s) |
| | Country/Region |
| | Sample size |
| | Job titles |
| | Abortion procedures and indications provided |
| Study Design | Qualitative/Quantitative/Mixed Methods |
| | Data collection method |
| | Sampling strategy |
| | Method of Analysis |
| Key findings (initial descriptive themes) | Abortion stigma within the community |
| | Experiences with/Views on Abortion legislation |
| | Challenging cases/aspects of abortion care |
| | Challenging interactions with patients* |
| | Challenging interactions with colleagues* |
| | Access to resources (incl. training, equipment, space, etc) |
| | Personal beliefs around abortion |
| | Influence of professional and personal experience |
| | Motivations to provide care |
| | Skills needed by abortion providers |
| | Supportive interactions within the workplace* |
| | Negative emotions linked to abortion care** |
| | Positive emotions linked to abortion care** |

*These three descriptive themes were initially labelled "Interactions in the workplace" but were expanded during the charting process.

**These two descriptive themes were initially labelled "Emotions linked to abortion care" but were expanded during the charting process.

## Stage 5—Collating, summarizing, and reporting the results

Data charted during the review were analyzed in two stages. The first stage was to finish the qualitative synthesis that had begun during the charting process, following guidance from Thomas and Harden [18]. As discussed in the previous section, we had devised 13 descriptive themes in the providers' experiences by the end of the charting process for the first selection of papers identified by our search. We then collated these descriptive themes to create analytical themes and sub-themes. This led to the creation of two broad themes in the providers' experiences. To write the results, we reviewed the charted passages and created narrative summaries of the extracted passages, paying attention to the convergences and divergences in the providers' experiences across the studies. This was an iterative process, and we revised the analytical themes and sub-themes throughout analysis and write-up. The second stage of the analysis was to conduct a descriptive numerical summary on the article reference details, study

contexts, and study designs to give an overview of the included studies. Once the first draft of the scoping review had been completed, we searched for newly published articles in the same electronic databases, and we looked for undiscovered articles in the reference lists of the articles included in the review. Data from these newly discovered articles were charted and their findings were congruent with the thematic structure that had been developed.

We also conducted a quality appraisal on the included studies using the *Mixed Methods Appraisal Tool* [19]. Following guidance from Levac *et al.* [14], we included a quality appraisal to aide in the interpretation of the findings and to ensure that results from methodologically weak studies are noted, strengthening the interpretations. Studies deemed to be of low methodological quality weren't removed from the review, rather their low quality is noted in the supplemental table summarizing each study (S2 Table) and the supplemental table that gives a detailed overview of the quality appraisal findings (S3 Table).

## Stage 6 –Consultation

A consultation meeting was conducted with providers in Ireland to discuss the thematic structure of the review. This meeting involved nine providers including GPs, obstetricians, and midwives. The providers agreed with the thematic structure, believing it to provide an appropriate and comprehensive overview of the potential challenges and facilitators to providing abortion care.

## Results

### Study selection, characteristics, & qualitative themes

The first search conducted in September 2020 identified a total of 11,402 citations. After removing duplicates, a total of 6,624 unique citations which were screened for inclusion. Of these, 78 relevant citations were included in the first draft of the scoping review. A further 14 newly published studies were found after conducting another search in the electronic databases in June 2022 and 14 studies were identified after we searched the reference lists of the included studies. This left a final pool of 106 studies in the review [20–125]. Fig 1 provides an overview of the search process. See Supplement 2 for a description of each of the included studies.

Of these 106 studies, 83 were qualitative, 15 were quantitative, and eight were mixed methods. They report on a total sample of 4,250 abortion care providers from 28 countries and six continents: North America ($n = 35$), Europe ($n = 31$), Africa ($n = 24$), Asia ($n = 9$), South America ($n = 6$), and Oceania ($n = 4$). See Fig 2 for the countries and number of studies per country. Regarding the professions included in the studies, 65 studies explored the experiences of nurses, 42 of midwives, and 64 of doctors (obstetrics and gynecology = 38, family medicine and general practice = 18, other or non-disclosed specialties = 31). Throughout the results, examples and quotes are presented alongside the country and job title of the provider who discussed them for context. The generic term *"provider"* is presented when there are more than three job titles for that example. Additionally, continents are given in some instances where many countries were named. A more detailed examination of how cultural values, such as societal views around abortion and the legal status of the procedure, in the different studied countries may influence providers' experiences is beyond the remit of this review. See S3 Table for a detailed overview of the quality appraisal and S4 for an overview of the legal status of abortion in the studied countries.

We found that the providers' experiences were related to one of two broad themes: (1) Providers' experiences with abortion stigma or (2) Providers' reflections on their abortion work. See S5 for supporting references for each theme.

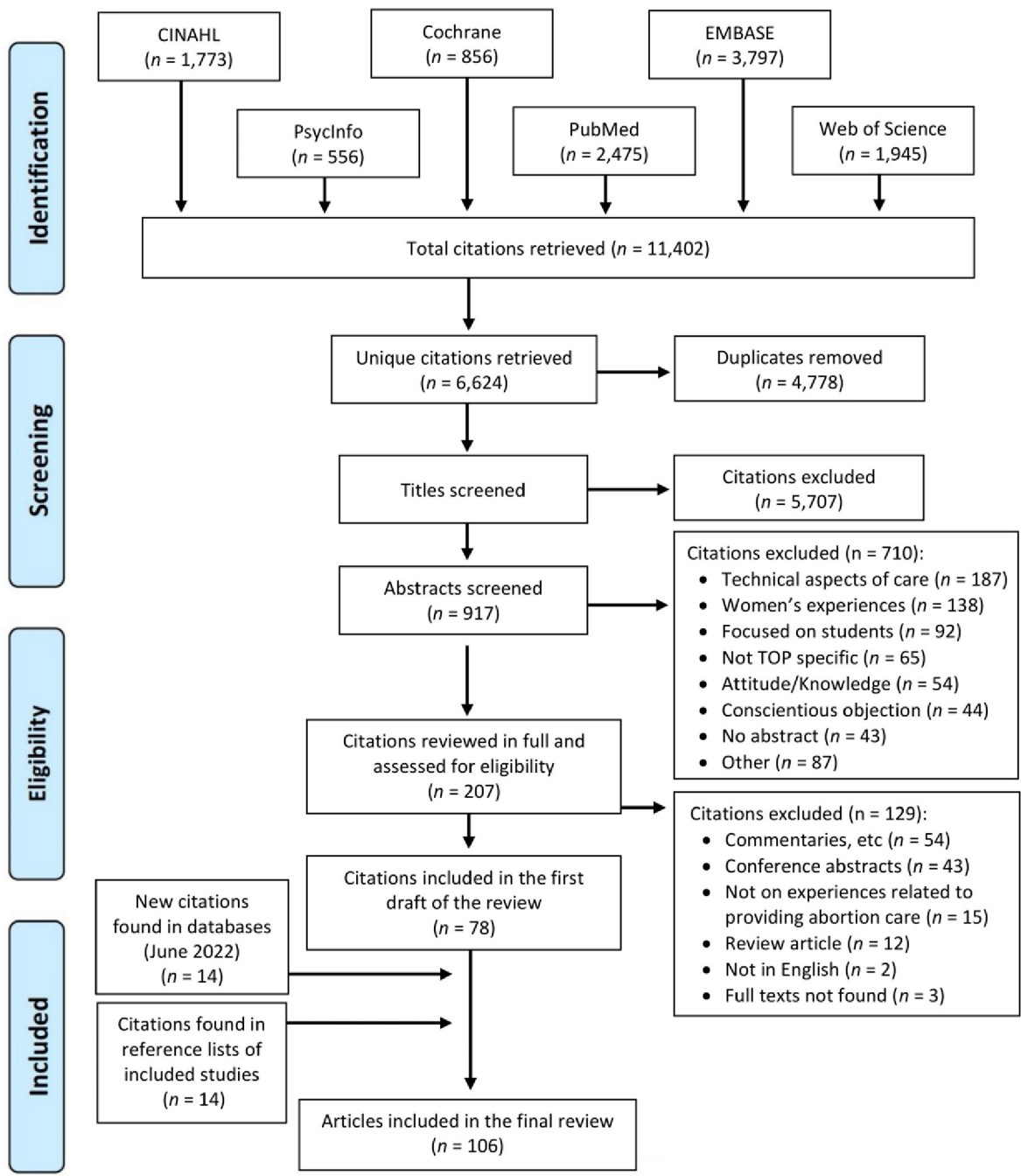

**Fig 1. Flow chart showing the study selection process for the scoping review on providers' experiences with abortion, adapted from the free chart designed by PRISMA [13].**

## Theme 1—Providers' experiences with abortion stigma

This theme explores providers' experiences which highlight the stigmatization of abortion care. Stigma is shown by experiences evident with society, and within the providers' communities and workplaces.

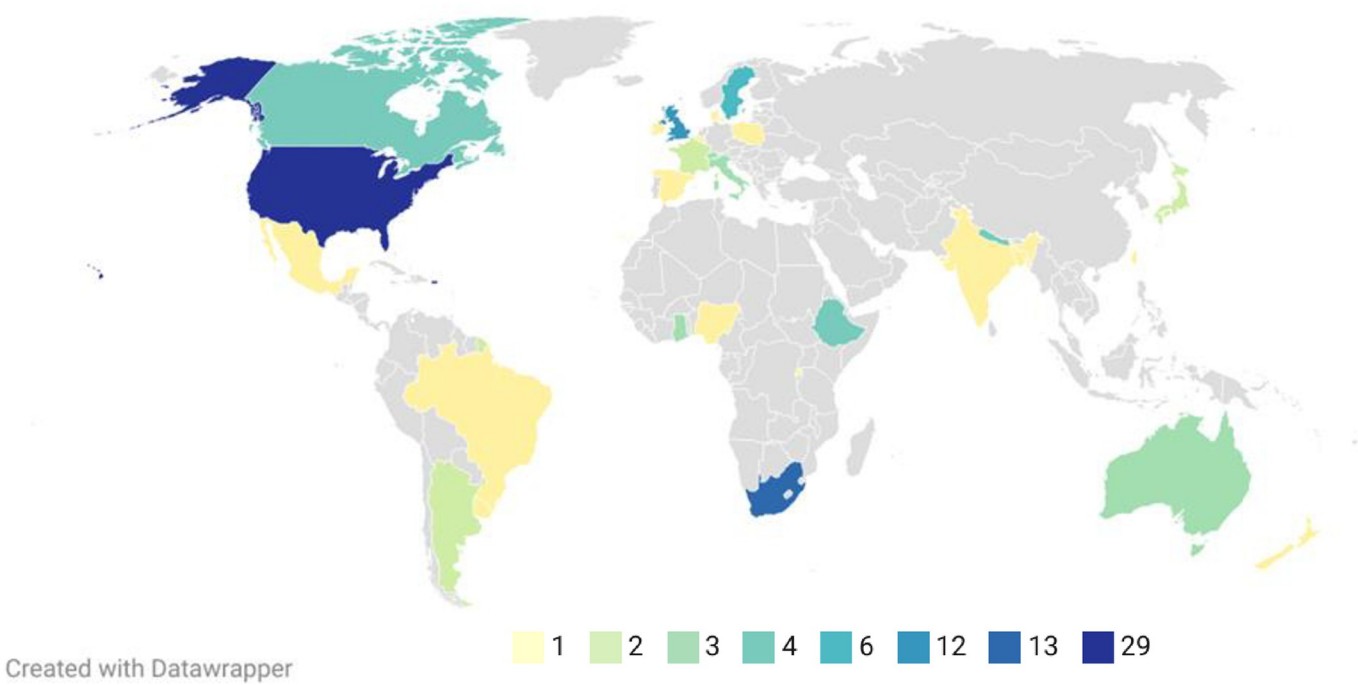

Created with Datawrapper

**Fig 2. Map showing the countries included in the scoping review.** The colors of each country indicate the number of studies included in the review for each country, according to the discrete categories at the bottom of the screen.

**Sub-theme 1—Within society.**    The stigmatization of abortion care was evident within society. Many providers discussed how inappropriate laws placed on their work can negatively affect public perceptions of abortion care, though some shared good experiences with laws. Many also felt that the media took an often-unbalanced view of abortion care.

*Laws governing abortion care.* Many described restrictive laws which impacted their ability to provide care, such as state-mandated counselling in the USA [20, 58, 74], mandatory waiting periods in Uruguay [52], and a restriction on abortion medications in Rwanda and the USA [32, 42]. Of state-mandated counselling in North Carolina, the USA, a provider said, *"It seems very clearly to be designed to shame women and guilt them out of deciding to have an abortion"* [99]. Restrictive laws may also impose undue institutional barriers, such as increases on the time or number of staff required to provide care [58, 74, 75]. Working under restrictive laws also opens providers to the threat of criminalization for completing their job [42, 60, 75, 94, 125]. For these reasons, some stated that restrictive laws stigmatize abortion and can damage the rapport between patient and provider, negatively impacting the standard of care [20, 42, 58, 70, 74].

Providers also said that abortion laws can be ambiguous in their interpretation. Providers in Ethiopia, Ghana, Mexico, and the USA voiced concerns in how to apply the law when providing abortion upon request, for example, deciding who is allowed to provide care, what paperwork is required, and what documentation does the patient need to present [28, 31, 64, 72, 92]. Obstetricians in Ireland and the UK also discussed anxiety in having to decide whether certain fetal abnormalities would be legally recognized as sufficient grounds for an abortion, which may leave providers at risk of criminalization if they unknowingly misinterpret the law [43, 66, 114].

While most studies highlighted the negative impacts of inappropriate laws, some noted positive experiences with abortions laws. Providers in Argentina said that consulting with medical

practitioners when writing the laws following liberalization helped to ensure they were appropriate [48], while providers in Bangladesh, Ethiopia, Nepal, and Uruguay noted that liberalizing abortion laws helped to reduce abortion stigma, which in turn reduced the numbers of unsafe abortions [28, 31, 52, 89].

*Anti-abortion sentiments in the media.* The media also highlighted negative sentiments on abortion care. Most providers in a study from the USA felt that the media rarely or never took a balanced view of abortion care [79], gynecologists working in Spain had been labelled as *"aborteros (back-street abortionists)"* who provide *"unsafe"* and *"dangerous"* procedures in the national media [53], and a midwife in the UK said that the media often present stories *"about little fetuses hanging on for dear life, on the back of bins"* [101]. Additionally, providers in Ireland and Mexico felt that the media were watching and waiting for a clinical error following liberalization in both nations [43, 92] and providers in India and the USA feared that clinical errors would become news headlines [51, 75].

**Sub-theme 2—Within the community.** The stigmatization of abortion care was also evident within local communities. Many described experiences with anti-abortion sentiments both within their wider communities and with family and friends. In response to these, providers discussed the challenge of disclosing their work.

*Anti-abortion sentiments in the community.* Experiences with anti-abortion sentiments were common within the local community. Providers across many countries in Africa, Australia, Europe, and North and South America described their discomfort in being known as an *"abortionist"* in their communities [30, 39, 42, 52, 56, 59, 64, 79, 82, 84, 92], with some recalling times they have been called *"assassins"* [92], *"murderers"* [53, 63, 64], or *"butchers"* [103]. Some recalled discrimination, such as a nurse in Scotland who said that discussing their abortion work is often *"a conversation stopper"* [39]. Quantitatively, as high as 50% of providers in two USA-based studies said that they have been verbally threatened or attacked because of their involvement in abortion care [56, 79], compared to 15% from a study in Ireland [27].

The studies also highlight that the frequency of anti-abortion experiences may be higher in certain communities. Providers working in a politically liberal state in the USA viewed themselves as *"lucky"* compared to peers working in more conservative areas, believing that they experienced fewer protests and perceived most people to be supportive of abortion care [99]. This is supported by providers in more conversative US states who discussed the high levels of stigma they face [20, 22, 58, 74]. High levels of stigma were also observed in conservative countries such as Ghana, Ethiopia, and South Africa [35, 45, 49, 64, 82]. Similarly, providers in rural areas of Australia, Canada, and Uruguay saying that they are not afforded the anonymity available to their colleagues in urban areas and experience stigma more often [52, 59, 84, 86]. Providers also acknowledged that many religious organizations oppose abortion [36, 39, 49, 51, 64, 94]. Providers in the USA expressed frustration that anti-abortion advocates often ground their views in religious beliefs, thus erasing the compatibility between holding spiritual beliefs and supporting abortion care [60].

*Anti-abortion sentiments from family and friends.* Experiences with anti-abortion sentiments were also described in interactions with family and friends. Providers in South Africa and the USA noted that they do not feel comfortable talking about their work at home as their partners think that abortion is *"distasteful"* [99, 117], an obstetrician in the UK said that their partner does not want their friends to know about their work [66], and providers in Ethiopia and the USA said that some family members or friends would never speak to them again if they knew about their work [33, 49, 94]. Close friends can also be a source of stigma, as a GP in Australia said:

*A few of my friends found it very difficult to deal with the thought of me doing [an abortion] and it took me a long time to actually tell them that I was doing it. . . they responded more negatively than I thought they would.*

[59]

*Hiding abortion work.* In response to anti-abortion sentiments within the community, many described the challenge in deciding whether to share their involvement in abortion care. Choosing not to disclose abortion work may protect providers and their families from abortion stigma. A GP in an Australian study noted that they do not advertise the abortion services in their clinic in fear of community backlash [59], physicians in Canada said that they *"fly under the radar"* and *"keep a low profile"* to avoid discrimination [84], and nurses in South Africa and the USA said that disclosing their work could damage existing friendships and make it difficult to create new ones [45, 94]. Providers also may choose to avoid disclosing their work to avoid *"danger talk,"* e.g. sharing difficult experiences of their work which may further stigmatize or limit access to abortion care [25, 60, 94, 99].

Others highlighted benefits in discussing their work. Most providers in two studies from the USA said they did not hide their work because of widespread support among friends and family [33, 99], while providers in the UK and USA said that disclosing their work allows them to combat stigma and misconceived perceptions about care [33, 39, 99], as well as avoid relationships with those who oppose abortion and identify like-minded individuals [33, 94, 99]. Additionally, sharing their work can help maintain psychological consistency as providers don't feel the need to hide their work despite supporting access to care [94]. Hearing how colleagues discuss how they disclose their work may encourage providers to share their own experiences [65].

**Sub-theme 3—Within the workplace.** Finally, the stigmatization of abortion care was evidenced by experiences within the providers' workplaces. These included negative experiences with their colleagues and the patients that access care. Despite this, many also highlighted positive experiences with their colleagues. Many also said that their services were inadequately resourced.

*Negative experiences with colleagues.* Like in the community, many providers noted experiences with anti-abortion sentiments from colleagues. Providers in Mexico, South Africa, the UK, and the USA said that some colleagues who did not support the abortion services created *"hostile"* work environments [39, 45, 63, 92, 100, 103, 106] and some labelled providers as *"serial killers"* [103] or *"baby killers"* [63, 88]. Other examples of anti-abortion sentiments included feelings of disapproval, rejection, and disrespect [43, 53, 56, 59, 78, 94, 103, 106], suspicion around complications [94], questioning of the providers skillset [56, 73], and rude interruptions during abortion consultations [45]. A nurse in Scotland elaborated on the experiences they face in the workplace:

*Everybody knows terminations happen [. . .] So I just wish it was a much more open, honest thing and that other staff would come and see we're not evil, we're not horrible and how much these patients need the support.*

[39]

As in the community, some chose to hide aspects of their work from their colleagues. This may be choosing not to tell people that they provide care [49] or with-holding certain aspects of care, like the fetal remains jar [94] or their own emotional reactions [45]. Providers may also hide aspects of their own lives from colleagues, such as their spirituality in fear that they

will be questioned on why they provide care [60]. Others may choose to reduce the number of patients they see or avoid providing care at all in fear of stigma and retaliation from colleagues [100].

While most supported their colleagues' right to conscientiously object from providing abortion care based on moral or ethical grounds, many felt that unclear *"opt-out"* procedures mean that staff who did not hold these objections refused to partake [37, 43, 92, 106]. They noted that high rates of conscientious objection can leave a small staff dealing with all abortion cases on top of their designated duties [24, 29, 34, 39, 43, 44, 52, 63, 66, 72, 78, 86, 92, 103, 104, 119]. There may also be little appreciation or renumeration for these additional duties [34, 43, 63, 92, 106]. Providers in Ireland, South Africa, the UK, and the USA said that procedures often need to be planned around those unwilling to participate [39, 43, 63, 100], while providers in community-based practice in Australia, Canada, and the USA said that high rates of conscientious objection creates issues when trying to organize external services, such as access to anesthesia, ultrasound, counselling, or a local pharmacy willing to dispense medication [32, 59, 84, 86]. In addition, some may *"conscientiously obstruct"* abortion by preventing women timely access to care [43, 53, 63].

Anti-abortion sentiments and high rates of conscientious objection left some feeling isolated, with nurses and midwives in South Africa and the USA feeling*"trapped"* in abortion care [63, 72, 73, 117], while obstetricians in Ireland said they had to run the services on their own [43]. Providers can also be isolated from one another. For example, providers in Ethiopia could not connect with colleagues as they felt unable to discuss their work due to stigma [49], while physicians in rural Australia, Canada, and the USA said that they lacked opportunities to connect with other providers [20, 22, 59, 84].

*Positive experiences with colleagues.* Many providers also discussed experiences with supportive colleagues, with one in the USA sharing that "*the most sustaining thing is probably the other people I work with. Because working in abortion, it draws really good people*" [99]. Providers highlighted the importance of being able to *"vent"* on the*"dangerous feelings"* and *"suffocating issues"* of abortion care and to have someone to *"bounce things off"* without fear of judgment [24, 25, 43, 45, 60, 61, 65, 94, 99, 101, 104, 115, 116, 121, 123], and colleagues who share similar experiences were identified as crucial [43, 61, 65, 101, 109, 113, 117]. Emotional support can occur through informal conversations between peers or through formal meetings with a peer group or a counsellor [24, 45, 60, 61, 65, 96, 99, 101, 103, 104, 109, 115, 117, 121, 123]. Despite this, however, some believed that access to peer support groups [81, 109, 115, 123], a non-biased psychologist or counsellor [45, 46, 61, 103, 109], and regular debriefing meetings [77, 103, 109, 121] were lacking. Attending local or national professional meetings may also help foster connections among staff [43, 84, 99]. Research on collegial support interventions, such as the *Provider's Share Workshop*, has also shown promising results in fostering connections between staff [60, 65, 94].

Colleagues can also offer support in the technical or procedural aspects of care. Providers in Canada, France, Ireland, Italy, and South Africa highlighted the importance of multi-disciplinary debriefings on technically or emotionally challenging aspects of care [20, 43, 45, 61, 71, 81, 103, 109, 121], while physicians in Canada and the USA discussed the importance of being able to attend continuing professional development and education events [20, 84]. As previously highlighted, where laws are ambiguous, obstetricians in Ireland and the UK highlighted the importance of being able to consult with colleagues on whether a fetal condition is compatible with the legal requirements [43, 66, 114].

Some also highlighted the benefits of working with an experienced colleague [59, 73, 112]. Nurses in the USA said that experienced colleagues can help new providers develop empathy [73], while providers in Ghana said that experienced mentors could validate the concerns of

newer providers and give advice [35]. Mentors may also be a source of emotional support. Experienced providers in Canada and the UK said they teach new providers strategies to deal with the difficult parts of care [81, 122], while nurses who worked as both providers and facility managers in South Africa said that they *"strive to make [their] co-workers feel at ease. I make them feel comfortable. We support each other"* [45].

*Anti-abortion sentiments from patients.* Some also noted anti-abortion sentiments from patients. These experiences may be indirect, such as patients in the UK and USA surprisingly saying *"you're so nice here"* and *"I wasn't expecting people to be as nice as they were"* [39, 99, 101]. Others may be more direct. Providers in separate studies from the USA said that patients sometimes express pro-life sentiments [94] or call the medical team *"killers"* when accessing care [60], while nurses and obstetricians in South Africa noted that some patients pressure them to provide care despite being over the legal gestation [45, 119]. Providers in the USA said that patients in a "*very difficult place*" may take their anger out on clinic workers and highlighted the importance of *"letting it go"* and not taking hostility from patients *"personally"* [111], while a doctor in the UK believed that *"the stigma [means] they think that the staff are going to be horrible to them"* [39].

*Under-resourced services.* Issues relating the human resources included a lack of staff willing to participate [22, 34, 41, 45, 63, 71, 78, 81, 88, 92, 98, 103, 106]. As previously noted, this may leave a minority of staff supporting the services and providers may be forced to take on an additional workload. Further, providers in Ghana, Italy, Mexico, Spain, and the USA highlighted the need for more providers to be trained in technical aspects such as surgical procedures or pain relief [42, 52–54, 64, 72, 88, 92, 115]. Providers in Mexico and South Africa said that more providers would allow the abortion workload to be shared [92, 103], while midwives in Italy and nurses in the USA said that providers should always work with a supportive colleague [71, 73].

In addition to human resources, providers across many studies noted the irregular supply of necessary equipment, such as instruments for vacuum aspiration [41, 63, 64, 78, 82, 92, 103], and medications and anesthetics [57, 63, 92, 115]. Providers in Canada, Nepal, South Africa, and the USA also noted that the space allocated for abortion care was often inappropriate [31, 41, 45, 57, 63, 72, 88, 89, 106, 115], while providers in Australia, Canada and Mexico said that it was difficult to secure rooms at all [59, 84, 92]. Providers also noted that it can be difficult to build a career in abortion care. Nurses in the USA said that opportunities to observe and participate in abortion during their training were lacking [73], while GPs in Australia and physicians in rural Canada highlighted that an irregular workload with periods of inactivity could make it difficult to acquire and retain skills [59, 84]. Nurses in the USA highlighted the lack of *"activities of legitimacy,"* such as professional meetings and societies, within abortion care, which help to foster career development and staff retention [73].

## Theme 2—Providers' reflections on their abortion work

This theme explores providers' reflections on their involvement in abortion care. Despite the widespread stigmatization of abortion, most reflected positively on their work. They emphasized support for the services, even if challenged by certain aspects, and believed that abortion is important work. Many also acknowledged the emotional impacts that providing abortion may present.

**Sub-theme 1—Providers' views on abortion.** Most providers said that they supported the right to bodily autonomy and self-determination. Many said that these views were informed by experiences which evidenced the need for a safe, legal abortion service, such as providers in Bangladesh, Ethiopia, Nepal, South Africa, Switzerland, and the UK who said that

by providing abortion, they are helping to reduce unsafe abortion, maternal mortality, and the occurrence of unwanted or abused children [28, 29, 31, 38, 49, 89, 90, 104, 118, 119]. Further, providers in the USA said that experiences within the workplace, such as hearing patients' stories, force providers to constantly refine, reinforce, and potentially reshape their beliefs around abortion to become more supportive [47].

Many also perceived a sense of duty to provide access to abortion care, such as providers in Canada, Italy, the UK, and Uruguay who said that only the patient should be allowed to decide what is right for them [39, 52, 66, 71, 101, 104, 115]. Affirming their sense of duty, nurses and physicians in the USA referred to their work as a *"calling"* and a *"passion"* [58], a midwife in Japan said that *"care during the delivery procedure is part of our professional responsibility. . . hence, I think that the midwife should play a part, even if this procedure means the end of a life"* [85], and a doctor in Nepal said that abortion care *"is a service to be given. We are the best persons to give them"* [89]. A minority, however, felt that their work was in contradiction to their professional duty to preserve life. Midwives and nurses in Japan, Switzerland, Taiwan, and the UK said that their typical responsibilities saw the creation of life while their abortion work sounded the *"the death knell"* [69, 98, 101, 120], while doctors in South Africa said that *"elective"* abortions were contrary to their oath to *"do no harm"* [119].

Regardless of personal views, most saw their main responsibility as providing non-biased information and empathetic care to all patients. Exhibiting commitment to these skills, providers in the USA affirmed that their *"role isn't to second-guess or question [the patient's] decision"* but to *"support that decision and to help women in choosing what method is best for them . . . without judgment or any negativity"* [58], while a midwife in Italy said that *"it is fundamental to be empathic with the woman . . . because some women really need psychological support"* [71]. Providers noted that experience may aid the development of these skills, as a nurse in the UK explained, *"I've been here long enough to judge how they're feeling. . . it vibrates off you. . . I can usually always win them round, even if they're not on my side at first"* [62]. Experiences within the providers' personal lives may also help in the development of these skills, with providers in South Africa, the UK, and the USA saying that experiences with pregnancy, parenthood, pregnancy loss, and abortion helped them to develop empathy [30, 47, 72, 99, 104, 123].

**Sub-theme 2—Challenging aspects of abortion care.** Despite generally positive views on abortion, many said that they were morally challenged by certain aspects of care. These included experiences with the fetus, differences between surgical and medical methods, providing care for cases that are medically indicated versus those that are not, and select cases where providers believed that abortion is being used as a method of contraception.

*The fetus.* Many discussed how contact with the fetal remains is a difficult aspect of care. Midwives in Japan noted sadness, confusion, and guilt when handling the remains [98], while nurses in the UK described the experience as *"undignified"*, *"horrible"*, and *"like a bereavement"* [104]. Increasing gestational age at the time of abortion was also a significant challenge, as a nurse in South Africa explained:

"*With first trimesters it is not so difficult or real, as the terminated pregnancy is more like a small blood clot or tissue but now with second trimesters. . . they have a human shape, something that we recognize and that suddenly makes it an awful lot more real*"

[88].

Procedures in the second and third trimesters may also be more technically and emotionally demanding than in the first trimester, with midwives and nurses in the UK describing second trimester abortion due to fetal abnormalities as *"going through the motions"* of a normal

delivery and how patients are often *"so upset because it's a perfectly formed little baby"* [107]. The short difference in time between later-gestation abortions and premature deliveries can also present a challenge, as highlighted by a midwife in Sweden, *"emotionally it is very hard . . . you save lives at 22–23 gestational weeks"* [96]. At later-gestations, the fetus may show signs of life at delivery, which many described as deeply distressing and the most challenging part of their work [43, 50, 71, 77, 83, 88, 98, 104, 109, 114, 115, 117, 121].

Contact with the fetus coupled with increasing gestational age and signs of life forced many to reflect on the unborn's right to life and their involvement in care [36, 46, 47, 50, 60, 62, 71, 85, 88, 98, 101, 104, 116, 117, 119, 120, 125]. Providers in Japan, Sweden, Taiwan, the UK, and the USA described strategies to reduce their moral discomfort, such as looking away or suppressing thoughts while handling the remains [60, 69, 77, 98, 101]. A provider in a study from the USA, however, believed it is important to acknowledge that abortion results in the loss of potential life and disagreed with prominent pro-choice discourse that the fetus is just *"a clump of cells"*:

> *You always see the pictures that anti-choice people protest with mangled fetuses and stuff. I had never really connected that to our work, so I was like, 'wow, this actually is, at that point, a little fetus.' Which is surprising to me, but it didn't change the way I feel about abortion at all . . . if anything, it probably made me more prochoice somehow, because I think it's really important to understand every aspect of abortion.*

[25]

*Medical vs surgical care.* Many made distinctions between providing care using medical or surgical methods. Staff generally viewed medical abortion as less technically and emotionally challenging, with providers in Uruguay saying that medical abortion requires less training than surgical methods [52], while an obstetrician in the USA shared surprise in *"how not emotional"* medical abortion can be [54]. Some also said that medical procedures are less morally challenging, as providers play a less active role and may not need to be present during expulsion of the fetus [50, 52, 62]. By contrast, involvement in surgical procedures, such as vacuum aspiration, dilation and curettage, or dilation and evacuation, ensure that providers have physical contact with fetus and may feel responsible for ending the pregnancy [47, 52–54, 63]. Some described surgical procedures as more emotionally demanding, with some referring to them as *"unpleasant"*, *"horrible"*, *"powerful"*, and *"traumatic"* [53, 54, 63]. Surgical abortion, however, may be preferable in certain circumstances. Midwives and nurses in the USA and Wales said that surgical procedures at later-gestations are much faster than medical induction and that the use of general anesthesia reduces the emotional weight of the procedure for patients and providers [107, 116].

*Value judgments about patients.* Many also believed it was easier to provide care if they perceived a legitimate reason for the abortion, for example, FFA, to protect maternal health, or in cases of rape or incest. Many differentiated between these and so-called *"elective"* or *"voluntary"* abortions [23, 47, 53, 66, 71, 89, 95]. A midwife in Sweden expressed sorrow at the termination of healthy fetuses in contrast to women struggling with fertility issues [77], providers in Ethiopia found it more difficult to justify their involvement if the motivation for abortion was not *"safeguarding the women's health"* [28, 29, 36], and a nurse in the USA said their colleagues are more empathetic *"if they feel that the client . . . had no control over [the decision to end the pregnancy]"* [72]. Abortion for non-lethal fetal abnormalities was also viewed negatively [71, 114, 115, 121], with midwives and obstetricians in France raising concerns over the place of disability in modern society [121].

Some also shared concerns that abortion can be used as a method of contraception, with repeat abortions and refusal of contraception being common examples. These were often referred to as *"unnecessary"* and less *"acceptable"* or *"worthy"* than medically indicated cases [26, 28, 47, 53, 68, 89, 91, 106, 117]. While these cases often evoked feelings of anger and frustration, some viewed them as a personal or systemic failure to provide effective education and access to contraception [26, 52, 68, 102]. A minority feared that discussing contraception during the abortion process could be judgmental or coercive [26, 68].

**Sub-theme 3—Providers' emotional responses.** Finally, providers highlighted the emotions they experience through their work. Many studies highlighted the difficult emotions related to abortion work, though many also highlighted positive emotions.

*Difficult emotions in abortion work.* Feelings of moral uncertainty were common and many providers in African, Asian, European, and North American studies expressed unease in the fact that abortion resulted in a loss of potential life. While supportive of patients' right to access care, one nurse in the UK reflected on the unavoidable reality of abortion care, *"I don't particularly like the outcome, that is that you're terminating all these fetuses"* [62]. For many, moral doubts were a source of confusion, sadness, and guilt, with a midwife in Japan saying, *"Why should this baby have life, but that baby cannot?. . . I don't know what human destiny is. . . (I have this) growing sense of guilt"* [98]. Some spoke about using prayer or reflection and evoked their religious moral norms of compassion and helping those in need to find comfort in the face of moral uncertainty, though this may be harder to do for *"elective"* procedures [36, 45, 47, 60, 82]. Others spoke about how they *"switched off"* during the procedure or while handling the fetal remains to protect themselves from moral doubts [97, 104, 111, 117]. Providers in a study from the USA, however, welcomed these challenges and reframed them as proof that they were still thoughtful and engaged with their work, with one physician saying that *"I am more guarded about not feeling, I never want to not feel"* [60].

*Positive emotions in abortion work.* Positive emotions encompassed pride, satisfaction, and happiness in providing care in an emotionally, physically, ethically, and morally challenging situation. A midwife in Italy said that *"receiving thanks at the end of the process is something wonderful, as it means that a woman has been completely satisfied during her difficulties"* [71], while nurses in the UK said that they take satisfaction in having *"done a good job"* in a *"challenging situation"* [104]. Providers in Nepal and the USA said that they take pride in their ability to care for patients and to make a positive impact on both their lives and society in general [58, 89], with a doctor in Nepal explaining, *"I feel very happy. . . every day I think I have relieved a woman of her burden"* [89]. In quantitative studies, most providers in two studies from the USA and one study from Ireland felt proud to be involved in abortion care [56, 79]. Reflecting on their abortion work, a physician in the USA said:

> *I like to get a big return on investment of my time and there are very few areas of medicine where one can spend fifteen minutes with a little skill, and a little passion, have such a dramatic impact on a patient, her family, and society at large. The good work that we do just ripples out in ever broadening circles.*

> [58]

## Discussion

While this is an expanding area of research, no review to date has sought to map the existing evidence on providers' experience of abortion care to explore what is known. By synthesizing the results of qualitative, quantitative, and mixed methods research from six continents, we

have discovered that providers across the world share similar experiences which highlight the stigmatization of abortion care. We also found that many providers feel pride and have constructed a positive identity about their abortion work, believing it to be important and necessary. In this section, we will reflect on how the findings of the scoping review relate to Ashforth and Kreiner's theoretical model on how staff in stigmatized professions create positive identities around their work [8, 10]. We will also reflect on the strengths and limitations of this review, as well as exploring recommendations for future research and practice.

### Evidence of abortion stigma

The included studies highlight the stigmatization of abortion care from the perspectives of providers. Firstly, abortion stigma can tangibly be seen in the often unsuitable laws that govern abortion [20, 42, 58, 70, 74] and in how the media portrays abortion [43, 53, 79, 92]. These stigmatized cultural values perpetuate negative connotations about abortion care by imposing criminalization on providers for doing their work [42, 43, 60, 66, 75, 94, 114, 125] and institutional demands such as mandated counselling [20, 58, 74], or by negatively impacting providers' rapport with their patients, potentially lowering the standard of care [20, 42, 58, 70, 74]. These negative connotations may include that qualified healthcare workers require governmental oversight to provide abortion care correctly, that abortion care is unhealthy or dangerous, and that abortion care is wrong. Evidence of these stigmatized cultural values was not limited to laws or the media and was also experienced in the providers' everyday interactions. Within their communities, many recalled experiences with anti-abortion sentiments from family members [33, 49, 94], friends [59], and others in their local communities [53, 63, 64, 92, 103]. Some recalled discrimination and harassment due to their involvement in care [27, 39, 56, 79].

Such experiences, or the threat thereof, may force providers to consider whether they will discuss their involvement in care. While hiding their abortion work may protect providers and their families from stigma, the need to hide their work may lead providers to internalize shame about their work, constitute a burden to remain quiet, lead to missed opportunities to connect with like-minded individuals, and may deter providers from accessing supports [60, 94, 99]. Additionally, choosing to hide abortion work may contribute to the *"legitimacy paradox"*, a cycle whereby choosing not to disclose abortion work means that negative stereotypes about service providers and the realities of abortion care are not challenged, stereotypes then lead to stigmatization, knowledge of stigma targeted at providers leads to fear of personally becoming a target, and fear of stigma and violence leads providers to hide their work [126]. On the other hand, talking about their work can be emotionally fatiguing, expose providers to anti-abortion sentiments, and can damage existing relationships and make it challenging to form new ones [94, 99].

Experiences which showed the stigmatization of abortion care were also common in the workplace, such as negative interactions with some of their colleagues and patients, as well as contending with the lack of resources allocated to the services. While providers may choose to hide their abortion work in their communities, it may not be possible to do so within the workplace. If the provider works within a non-supportive or even *"hostile"* environment, they may constantly be exposed to anti-abortion sentiments [39, 45, 63, 92, 100, 103]. Further, some described feeling *"trapped"* and isolated within their role [63, 72, 73, 117] and noted the lack of opportunities to gain skill and progress their career [59, 73, 84]. This marginalization is in-keeping with Ashforth and Kreiner's model of stigmatized work [8–10, 127]. Repeated exposure to such experiences can be discouraging and may create moral doubts about their

involvement. Indeed, some even noted that they have been asked *"How can you do it?" [72]*, a trademark of stigmatized work [8].

Indeed, the providers themselves were not immune to abortion stigma. Many expressed stigmatized views and value judgments about some patients who choose to access care. This was particularly true for patients who accessed the services multiple times or did not use contraception [26, 28, 47, 53, 68, 89, 91, 106, 117], or who did not have a medical indication for accessing care [23, 47, 53, 66, 71, 89, 95]. While most supported access to care, these value judgments showed that many providers viewed some cases of abortion care as more or less acceptable than others. Beyond the providers' direct experiences, these views also perpetuate negative connotations that abortion is wrong, and providers may expose the women accessing care to anti-abortion sentiments, as has previously been found [128]. While some providers shared mostly negative reflections on their abortion work [69, 98, 101, 119, 120], most were supportive of their work. It is important that providers be supported in their work and receive resources, such as Value Clarifications workshops, that can help them to reflect on and discuss the moral challenges that they experience when providing abortion care [129].

## Creating a positive identity around abortion work

While the providers described their experiences that highlight abortion stigma, most also emphasised the positive experiences they perceive in their abortion work. As highlighted by Ashforth and Kreiner, though employees in stigmatized work may face challenges, they *"do not tend to suffer from low occupational prestige"* given *"that individuals seek to enhance their self-esteem through their social identities"* [8]. Therefore, this is in-keeping with the findings of our review that providers' can negate the stigma ascribed to their work and find meaning and pride when providing abortion care. In this section, we will reflect on how this may occur, ordering our results by Ashforth and Kreiner's theoretical model of stigmatized work [8, 10].

Firstly, Ashforth and Kreiner theorize that strong workgroup cultures among staff in stigmatized work promotes the use of social weighting processes and occupational ideologies, as staff collectively work together to challenge the stigmatized nature of their work and create a shared positive work role identity [8, 10]. These strategies will be discussed in more detail later. First, the results of this review support that many providers experienced a strong workgroup culture. Many reflected positively on the support they receive from their colleagues, such as being able to *"vent"* on the *"dangerous feelings"* and *"suffocating issues"* of abortion care without fear of judgment [24, 25, 43, 45, 60, 61, 65, 94, 99, 101, 104, 115, 116, 121, 123], as well as receiving emotional support through informal conversations or formal meetings [24, 45, 60, 61, 65, 96, 99, 101, 103, 104, 109, 115, 117, 121, 123]. While support was noted as a key importance, however, providers in some studies noted that access to resources such as support groups [81, 109, 115, 123] or a non-biased psychologist or counsellor [45, 46, 61, 103, 109] were insufficient. Additionally, while support from fellow providers was commonly discussed, support from non-providing colleagues is also important as management and peers alike have a role in normalizing the place of abortion within the workplace. It is vitally important that providers be supported given the stigmatized nature and moral challenges many experienced. A recent review from the Royal College of Obstetricians and Gynecologists reviewed evidence on supports available to providers and found that many can be effective in reducing the impact of abortion stigma [129]. Further strategies should be employed to improve peer support among providers and to implement these resources within healthcare systems.

As highlighted in the last section, Ashforth and Kreiner theorise that a strong workgroup culture promotes the use of social weighting techniques and occupational ideologies that help to negate the stigma and support staff in building a positive work role identity [8, 10]. Social

weighting processes include condemning the condemners, supporting the supports, and selective social comparisons, while occupational ideologies include reframing, recalibrating, and refocusing. The providers condemned those who opposed the provision of safe abortion care, such as those who protest abortion care [25, 99], the media when they perpetuate stigma [101], governments who endorse unsuitable legislation [99], or simply those who respond with stigma to their work. They also spoke positively of those who support access to liberalised care, as well as their providing colleagues. Many of the providers also implicitly sought positive social comparisons with their non-providing colleagues, by highlighting that their conscientious provision of abortion care is a professional duty or a moral calling while noting the challenges that high levels of conscientious objection can have on the provision of safe care [43, 53, 63].

Additionally, the occupational ideologies were common among the studies. The providers reframed the nature of abortion care from stigmatized work into a central tenet of their professional duty and/or a moral calling [58]. Many highlighted the nature of providing abortion care as a privilege with emotional weight, as saving women from unsafe abortion, as compassionate care to woman with pregnancies affected by fetal abnormalities, and as important work that has a positive impact on society [28, 29, 31, 38, 49, 89, 90, 104, 118, 119]. Providers in the USA even reframed difficult aspects of abortion care, such as handling the fetal remains, as evidence that they were still thoughtful and engaged with their work [60]. Affirming that abortion is socially important and necessary is in keeping with the model of stigmatized work, in that having "necessity shield" helps employees to construct a positive work role identity and resist stigma as they feel that their work is vital for society [8, 10].

Many also detailed how they recalibrate and refocused aspects of their work. When asked about their involvement in care, many emphasised their support for and commitment to provide care for cases that they believed to be "worthy" abortions, while minimising aspects of care which they or society perceived to be difficult or"less worthy," such as the non-medically indicated cases that challenged their support of the services [23, 47, 53, 66, 71, 89, 95]. Providers also recalibrated and refocused their work by underscoring empathy as a required skill [58, 71]. By emphasising the importance of psychological care in abortion, the providers were able to mitigate the physical aspects of care that for many were a source of moral discomfort, e.g. proximity to the fetal remains and involvement in later-gestation or surgical care. Many also recalibrated and refocused their perceived duty to provide care to their patients as paramount to their own beliefs and values [39, 52, 66, 71, 101, 104, 115], which may help providers to create moral distance from their patients' decisions to access care and from any perceived duty to care for the fetus. By reframing, recalibrating, and refocusing their abortion work in these ways, the providers diminished the challenging aspects of their work which created moral uncertainty, the most damaging aspect of stigmatized work [10].

The collective use of these strategies, as Ashforth and Kreiner say, helps to *"externalise or attribute the dirty work stigma to the ignorance or malevolence of outsiders"* [8]. Indeed, the results of the scoping review found that many providers reflected positively on their involvement in abortion care and generally did not suffer from low occupational prestige. Despite the abundant challenges they faced, most supported access to liberalised abortion care, perceived a duty to facilitate access to care, and held pride and satisfaction in their ability to provide care. This also supports a dialectic that abortion care can be an emotionally and morally challenging and a necessary and rewarding service to provide [5, 130, 131].

## Strengths and limitations

A key strength of this review was the use of the Levac, et al. [14] and PRISMA-ScR [13] guidelines, which helped to increase the rigor and replicability of the review. The inclusion of a

quality appraisal tool also aides in the meaningful interpretation of our findings [19]. Another strength is that the review compiles the experiences of 4,250 providers from 28 countries and six continents. We also, however, acknowledge that the geographic spread of the studies can be considered as a limitation. Two thirds of the included studies come from North America ($n = 35$) and Europe ($n = 31$) alone. While Africa appears well represented ($n = 24$), over half of the studies are from South Africa ($n = 13$). Further, studies on providers' experiences in Asia ($n = 9$), South America ($n = 6$), and Oceania ($n = 4$) are lacking. We acknowledge that studies, particularly those from Asia and South America, may have been published in native languages and given the constraints of this review, we could only include citations published in English. We advise that further research be conducted throughout Africa, Australia, Asia, and South America to explore providers' experiences. Additionally, a detailed cross-cultural examination of providers' experiences was not possible in the current review given time constraints and the small number of studies for many of the included countries. We acknowledge that the specific cultural values of each of the studied countries may uniquely affect the experiences of the providers working within them, and we recommend that future research explore this in more detail, perhaps by replicating this review and focusing on research from a specific country or region or focusing on specific aspects of the providers' experiences, e.g., their experiences with abortion training programs. In addition, further research may look at temporal differences between countries to explore how legislative changes may impact providers' experiences with care. An example may be a review looking specifically at providers' experiences with care in the USA following the repeal of Roe v Wade in 2022.

Secondly, while we searched for literature using a variety of methods (i.e., systematic searches in six electronic databases, citation indexing of the included papers, and hand searching relevant journals), we acknowledge that some papers may have been missed. We do, however, think that the impact of this may be minimal given the large number of studies included in the review. Thirdly, we note that Levac et al. recommend that review authors consult about the results scoping review with people who have lived experience of the topic under study to validate the results. Due to time constraints of this project, only one consultation meeting was conducted with nine providers working in Ireland. Though this group agreed with the findings of the review, we acknowledge that this is not representative of all providers included in the review. Finally, due to the high number of citations identified by our search and the time constraints, we decided only to include peer-reviewed, original research articles. This excluded sources such as grey literature, editorials, newspaper articles, and books which may have given further context to our findings.

## Conclusions and recommendations

This is the first review to map the existing evidence on the experiences of abortion care providers. Throughout the paper, we have discussed the known challenges of providing abortion care and have reflected on the factors which help providers to construct positive social identities around their work. We acknowledge that certain experiences may not be captured within the literature or that the experiences we have detailed may need further exploration. In addition to the future research studies that we have recommended, we encourage international providers, researchers, and advocates to build on the findings of our review and to contribute to this area of research as they see appropriate.

Considering our results, we also believe that certain practical actions can be taken to improve providers' experiences. Regarding legislation and the allocation of appropriate resources, political engagement could give providers the agency to highlight the challenges they face and to suggest improvements that could benefit providers and patients alike.

Regarding support in the workplace, formal and informal systems which foster connections amongst providers could be designed and implemented, such as education in desired areas, debriefing meetings, or peer support groups. National meetings could also be utilized to foster connections amongst providers. Specific support may also be beneficial for those who experience moral doubts or negative emotions related to their abortion work, such as one on one time with a counsellor or participation in values clarification workshops. Supports may also be used to combat stigma within the workplace by encouraging non-providers to explore their owns values and listen to those of providers, and by combatting misinformation about the services. More detailed examples of such resources can be found in a review compiled by the Royal College of Obstetricians and Gynecologists [129].

## Supporting information

**S1 Table. Preferred Reporting Items for Systematic reviews and Meta-Analyses extension for Scoping Reviews (PRISMA-ScR) checklist for the scoping review on providers' experiences with abortion care.**
(DOCX)

**S2 Table. Overview of the studies included in the scoping review on providers' experiences with abortion care.**
(DOCX)

**S3 Table. Overview of the quality appraisal on the studies included in the scoping review on providers' experiences with abortion care.**
(DOCX)

**S4 Table. Overview of the laws governing abortion care in each of the countries studied in the scoping review on providers' experiences with abortion care.**
(DOCX)

**S5 Table. Overview of descriptive and analytical themes devised as part of the scoping review on providers' experiences with abortion care with supporting references.**
(DOCX)

## Acknowledgments

We would like to thank Professor Patricia Fitzpatrick and Professor Walter Cullen for their advice at the inception of this review. This work was supported by the National Maternity Hospital, Dublin, Ireland in fulfillment of an academic qualification to be completed by the lead author. The funding organization was not involved in designing and conducting this scoping review, in writing this article, or in deciding to submit this article for publication.

## Author Contributions

**Conceptualization:** B. Dempsey, M. F. Higgins.

**Data curation:** B. Dempsey, S. Callaghan, M. F. Higgins.

**Formal analysis:** B. Dempsey, S. Callaghan, M. F. Higgins.

**Investigation:** B. Dempsey, S. Callaghan, M. F. Higgins.

**Methodology:** B. Dempsey, S. Callaghan, M. F. Higgins.

**Project administration:** B. Dempsey.

**Resources:** B. Dempsey.

**Supervision:** S. Callaghan, M. F. Higgins.

**Validation:** B. Dempsey, S. Callaghan, M. F. Higgins.

**Visualization:** B. Dempsey, S. Callaghan, M. F. Higgins.

**Writing – original draft:** B. Dempsey, M. F. Higgins.

**Writing – review & editing:** B. Dempsey, S. Callaghan, M. F. Higgins.

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
