## [Decision Letter · Decision Letter 0]

30 Jun 2023

PONE-D-22-24285Providers’ experiences with abortion care: A scoping reviewPLOS ONE

Dear Dr. Dempsey,

Thank you for submitting your manuscript to PLOS ONE. After careful consideration, we feel that it has merit but does not fully meet PLOS ONE’s publication criteria as it currently stands. Therefore, we invite you to submit a revised version of the manuscript that addresses the points raised during the review process.

I would like to apologise for the excessive delay in returning this review to you - I was only assigned late in the review process, and it proved challenging to find reviewers willing and able to consider your paper. As you will see, the reviews are quite different, with reviewer 2 highlighting a number of organisational issues of the paper that I believe would be useful to address, to improve the readers' understanding of your work.   Please submit your revised manuscript by Aug 14 2023 11:59PM. If you will need more time than this to complete your revisions, please reply to this message or contact the journal office at plosone@plos.org. Please include the following items when submitting your revised manuscript:A rebuttal letter that responds to each point raised by the academic editor and reviewer(s). You should upload this letter as a separate file labeled 'Response to Reviewers'.A marked-up copy of your manuscript that highlights changes made to the original version. You should upload this as a separate file labeled 'Revised Manuscript with Track Changes'.An unmarked version of your revised paper without tracked changes. You should upload this as a separate file labeled 'Manuscript'.

We look forward to receiving your revised manuscript.

Kind regards,

Rafael Van den Bergh

Academic Editor

PLOS ONE

Journal Requirements:

"Two authors (BD and MFH) are involved in other research studies exploring providers' experiences with abortion care in the Republic of Ireland."

4. We note that Figure 2 in your submission contain map images which may be copyrighted. All PLOS content is published under the Creative Commons Attribution License (CC BY 4.0), which means that the manuscript, images, and Supporting Information files will be freely available online, and any third party is permitted to access, download, copy, distribute, and use these materials in any way, even commercially, with proper attribution. For these reasons, we cannot publish previously copyrighted maps or satellite images created using proprietary data, such as Google software (Google Maps, Street View, and Earth). For more information, see our copyright guidelines: http://journals.plos.org/plosone/s/licenses-and-copyright.

Additional Editor Comments:

I commend the authors for their work on this review, and in particular the scope they have taken when conducting their review. I appreciate the inclusive approach of considering qualitative, quantitative and mixed methods studies. I would however suggest, in addition to the comments raised by reviewer 2, that the synthesis of the quantitative papers be considered in more detail. In particular, I would suggest that the authors summarise which were the main outcomes that were considered in the different quantitative studies, and report on the designs of these studies.

Reviewers' comments:

Reviewer's Responses to Questions

**Comments to the Author**

1. Is the manuscript technically sound, and do the data support the conclusions?

Reviewer #1: Yes

Reviewer #2: Partly

2. Has the statistical analysis been performed appropriately and rigorously? 

Reviewer #1: N/A

Reviewer #2: N/A

3. Have the authors made all data underlying the findings in their manuscript fully available?

Reviewer #1: Yes

Reviewer #2: Yes

4. Is the manuscript presented in an intelligible fashion and written in standard English?

Reviewer #1: Yes

Reviewer #2: Yes

5. Review Comments to the Author

Reviewer #1: The manuscript was a scoping review on providers experience of abortion. The manuscript was well written and the authors provided details on the methods to enable replication of the study. The results were also well written. I therefore recommend the manuscript to be accepted

Reviewer #2: 1. General Points

Overall, I found the focus of the topic and findings to be relevant and important. It was clear from reading this draft manuscript that the lead and co-authors have undertaken a comprehensive exploration of the studies selected for review. The complement of draft manuscript, tables and figures and supplementary information indicate a fastidious and meticulous approach to this work, which should be commended.

The draft manuscript would benefit from a number of revisions from both major and minor perspectives to transform it into a digestible and relevant article for intended and interested audiences. I have tried to include examples linked to the manuscript to provide the authors with a more tangible illustration of where revisions might improve the manuscript. These examples are not exhaustive, and further examples in the rest of the manuscript will also warrant attention.

2. Major Points

Results

Structure of themes/sub-themes: I feel that the existing structure of the Results presents an overly complex picture of the core findings of this research, and would benefit from the themes/sub-themes being revisited. With a total of 42 sub-sections in the Results, it is challenging to follow the logic of the findings as they are presented. Additionally, I note that the structure of the themes presented in Figure 3, the ordering of the manuscripts Results section and the further details provided in Table 5A of the supplemental information do not follow corresponding structures.

I would suggest the authors review the sub-themes under the two main themes and consider merging or consolidating at the sub-theme level. This is perhaps most relevant to section 2 of the Results, e.g., merging “3.3.1.1 Personal support for access to abortion” with “3.3.2 Sub-Theme 2 – Motivations to Provide Abortion Care” would consolidate views and motivations into a single theme. This would enable section “3.3.1.2” to become a separate sub-theme focused entirely on challenges, and then closing with sub-theme 3 on the emotional responses.

Additionally, I would recommend re-thinking the ordering of the results presented, e.g., in section 1 of the Results, the authors present first the community, then move to the wider framework of how the legislative landscape impacts, then back down to the workplace (including the patient level) perspective. Perhaps starting at the widest legal and administrative context, then moving to the community/media/personal circles and then on to the workplace environment would help the reader move between the locations identified in the findings. This approach has been adopted in section 4.1 of the Discussion where it works well.

Role of geographic locations: Throughout the Results there is limited analysis of how the geographical locations of the studies impact on the results presented. Numerous paragraphs include multiple references to the countries from which the studies were conducted, however the locations do not seem to have much of a bearing on the findings presented. E.g., 1: “Section 3.2.3.4.2 Access to training” covers a range of training topics (including to increase the number of abortion providers available, for technical skill development, to provide psychological support, for improved after-care, and to navigate legislation or ethical and moral issues). However, no links between the types of training identified and the locations the paragraph refers to are made. E.g., 2: “3.3.3.1 Difficult emotions in abortion work” flips from geographical regions to individual countries and back again, and it is unclear why regions or individual countries are being identified to support the findings presented.

Use of precise and accurate language: Throughout the Results, please ensure that findings are precisely and accurately presented. On occasion, the authors selection of specific language conveys different meaning to the findings than may actually have been identified. e.g: “Difficult emotions were also linked to restrictive policies and practices in abortion care, such as a nurse in Australia who felt the need to break the rules to provide optimum care: “I was scared, wasn't sleeping, hardly eating, started smoking… but what sustained me was that I knew I was doing the right thing and if I walked away from this and did nothing, then that would be a lot worse. I couldn't do that.”” It is unclear to me how the nurse in Australia has broken any rules.

Use of quotes: I greatly appreciated the use of direct quotes from abortion providers both in text and separate paragraphs throughout the Results section, striking a good balance in the ratio of text to quotes. They provide appropriate emphasis and illustration of findings. However, with the longer separate paragraphs, I would advise the authors to consider using a wider range of supporting quotes than those currently selected. Of the 12 longer quotes included, 10 are from the USA and 2 from Australia. Whilst there is a concentration of research in the USA, it is important to draw in voices from across the abortion provider landscape.

Length: Whilst there is no word limit for Plos ONE, the Results section is lengthy and challenging to digest. I suggest being more efficient with words and succinct in your communications. A reduced word count by approximately one third (to bring the results closer to 4-5,000 words) would improve readability considerably.

Results/Discussion distinction: There are occasions throughout the Results where the authors stray into the territory of the Discussion. Detailed review of the Results section would help to pull out the various sentences and sections of paragraphs that extend beyond the scope of Results and move into building a more comprehensive and meaningful Discussion. E.g. “3.3.1.1 Personal support for access to abortion. […] This moral justification about the need for abortion allows providers to legitimize their work and reduce moral doubts about their own involvement. It may also help to bolster providers against the challenges they may face.”

Discussion

The idea to explore resisting stigma in the Discussion section through “reframing, recalibrating, and refocusing” feels logical given the range of findings presenting in the Results section. However, the manuscript’s current engagement and exploration of this framing is insufficient. I would expect to see a much more comprehensive exploration of all three tenants to resisting stigma than currently presented.

For example, if reframing is about transforming abortion from stigmatized to honourable work, I would expect a more nuanced discussion on how communication plays a key role, including the careful curation by abortion providers around the narrative they present to others on their work, such as decisions to withhold specific information, careful selection of who to engage with and why. Furthermore, the work to reduce legal ambiguity, address stigma, legitimise abortion and reduce uptake of unsafe abortion wasn’t discussed. Whilst a supportive environment was built into the reframing component of the Discussion, it seemed to fall short of building the connection to honourable work, whereby “good people” (line 379) work in the sector.

I would encourage the authors to be more precise in their language choices when discussing how abortion providers navigate resisting stigma. For example, “By emphasising the importance of psychological care in abortion, they reduced the importance of the physical aspects of care which for many were a source of moral uncertainty…”. I think the intention here is to convey that placing emphasis on phycological care helps abortion providers to mitigate the emotional discomfort or moral uncertainty they face when providing specific physical care. However, by stating it “reduced the importance” implies that physical care in its entirety is secondary to psychological care.

Conclusions and Recommendations

Without a fully explored Discussion section, the Conclusions and Recommendations section is disconnected from the rest of the manuscript, introducing ideas that have not been developed elsewhere. Additionally it moves between practical and research recommendations multiple times and would benefit from a more logical structure.

3. Minor Points

Editing/Length: As mentioned elsewhere, whilst there is no word limit for Plos ONE, the current length of the Results makes it challenging to digest and the Discussion would benefit from expansion. Additionally, further fine editing to be more precise with language would help to convey more accurately the important points the authors wish to make.

The term “managing disclosure” within the field of sexual and reproductive health and rights is more usually reserved for aspects of patient care rather than professional practice (e.g. HIV status, or undertaking an abortion). I would therefore recommend re-considering the use of “managing disclosure” in your manuscript whereby you refer to an abortion providers declaration on the nature of their work.

Figures/Tables: All figures and tables would benefit from more precise and appropriate labelling to communicate to the audience the nature of the content, e.g. “Figure 3: Overview of themes and sub-themes for the scoping review.” This figure description currently indicates that the themes and sub-themes were pre-determined (“for”) rather than derived through the process of the research.

For Figure 2 specifically, rather than a discrete scale to indicate number of studies conducted in countries, a sliding scale would be preferable and more in line with data visualisation standards. The current figure capture does not adequately indicate what is presented.

Methods/2.2.1 Eligibility Criteria: The authors note that no time bound limits were imposed on inclusion criteria for studies. However, there have been both legislative changes as well as innovation in models of care for abortion (e.g. self-administration). These shifting landscapes in both the policy and practice of abortion care will inevitably have had a bearing on the experiences of abortion providers in those contexts pre- and post-introduction, which may no longer be reflective of the current situation. I would suggest that the authors include a stronger rationale for not time bounding the research and acknowledge the challenge it poses in the Limitations.

Methods/2.3 Stage 3 - Study Selection: The process and roles played by the three co-authors in the inclusion/exclusion of studies for review isn’t particularly clear. I suggest a more explicit paragraph in section “2.3 Stage 3 – Study Selection”. Figure 1 would benefit from the inclusion of the total number of studies generated, and the number of duplicates removed.

Limitations: I note from the Methods and Materials section that a stage for consultation was undertaken and that this was with abortion providers in the Republic of Ireland. I would ask the authors to reflect on whether the composition of the consultation group should this be considered a limitation. It is unlikely individuals in the consultation group will be representative of either the geographical spread of studies included in the review, or the abortion providers represented in those studies. They will offer a particular set of perspectives that will more likely reflect the Republic of Ireland, which are unlikely to be generalisable.

6. PLOS authors have the option to publish the peer review history of their article (what does this mean?). If published, this will include your full peer review and any attached files.

Reviewer #1: No

Reviewer #2: No

---

## [Author Response · Author response to Decision Letter 0]

31 Aug 2023

Journal Requirements:

We have revised the manuscript in line with the guidelines in these documents. 

"Two authors (BD and MFH) are involved in other research studies exploring providers' experiences with abortion care in the Republic of Ireland."

Thank you. I have updated this information in the submission portal and cover letter. 

A mistake on my behalf. I have made this edit when resubmitting the article.

4. We note that Figure 2 in your submission contain map images which may be copyrighted. All PLOS content is published under the Creative Commons Attribution License (CC BY 4.0), which means that the manuscript, images, and Supporting Information files will be freely available online, and any third party is permitted to access, download, copy, distribute, and use these materials in any way, even commercially, with proper attribution. For these reasons, we cannot publish previously copyrighted maps or satellite images created using proprietary data, such as Google software (Google Maps, Street View, and Earth). 

For more information, see our copyright guidelines: http://journals.plos.org/plosone/s/licenses-and-copyright.

In reviewing the materials used to produce our figures, both are copyright free and eligible for inclusion within our article. For Figure 1, we note that the PRISMA Flow Diagram is a commonly used and published figures within reviews. We have included a reference to the PRISMA-ScR guidelines article to provide credit to the original authors. For Figure 2, we have included a link to the Datawrapper website that supports this conclusion:

https://www.datawrapper.de/faq#can-i-use-my-charts-in-a-printed-publication

We acknowledge that while both are free to use and compliant with the CC BY license, we need to correctly credit the original creators in the figure captions. We have updated the captions for both figures to clarify this.

See response above, with thanks

Additional Editor Comments:

I commend the authors for their work on this review, and in particular the scope they have taken when conducting their review. I appreciate the inclusive approach of considering qualitative, quantitative and mixed methods studies. I would however suggest, in addition to the comments raised by reviewer 2, that the synthesis of the quantitative papers be considered in more detail. In particular, I would suggest that the authors summarise which were the main outcomes that were considered in the different quantitative studies, and report on the designs of these studies.

Thank you for your positive comments on our work. It was certainly a large undertaking, and we are happy that you and the reviewers see the value in our effort. We have included more on this in our paper. In summary, we did not specify exact quantitative outcomes that were to be considered, rather we carefully charted information that was relevant to providers’ experiences. We extracted and referenced relevant passages from the articles and then coded the written passages for inclusion within our descriptive themes. 

5. Review Comments to the Author

Reviewer #1:

The manuscript was a scoping review on providers experience of abortion. The manuscript was well written and the authors provided details on the methods to enable replication of the study. The results were also well written. I therefore recommend the manuscript to be accepted

Thank you for your time in reviewing our article and for your support.

Reviewer #2:

1. General Points

Overall, I found the focus of the topic and findings to be relevant and important. It was clear from reading this draft manuscript that the lead and co-authors have undertaken a comprehensive exploration of the studies selected for review. The complement of draft manuscript, tables and figures and supplementary information indicate a fastidious and meticulous approach to this work, which should be commended.

The draft manuscript would benefit from a number of revisions from both major and minor perspectives to transform it into a digestible and relevant article for intended and interested audiences. I have tried to include examples linked to the manuscript to provide the authors with a more tangible illustration of where revisions might improve the manuscript. These examples are not exhaustive, and further examples in the rest of the manuscript will also warrant attention.

Thank you for your support for our work and for your time and expertise in reviewing our article. We are glad that you see the value in our article and see the work that has gone into producing this report. Thank you for the examples you have given throughout. We think that your suggestions have helped to improve the quality of the manuscript immensely and we hope that both you and the editor agree.

2. Major Points

Results

Structure of themes/sub-themes: I feel that the existing structure of the Results presents an overly complex picture of the core findings of this research, and would benefit from the themes/sub-themes being revisited. With a total of 42 sub-sections in the Results, it is challenging to follow the logic of the findings as they are presented. Additionally, I note that the structure of the themes presented in Figure 3, the ordering of the manuscripts Results section and the further details provided in Table 5A of the supplemental information do not follow corresponding structures.

I would suggest the authors review the sub-themes under the two main themes and consider merging or consolidating at the sub-theme level. This is perhaps most relevant to section 2 of the Results, e.g., merging “3.3.1.1 Personal support for access to abortion” with “3.3.2 Sub-Theme 2 – Motivations to Provide Abortion Care” would consolidate views and motivations into a single theme. This would enable section “3.3.1.2” to become a separate sub-theme focused entirely on challenges, and then closing with sub-theme 3 on the emotional responses.

Additionally, I would recommend re-thinking the ordering of the results presented, e.g., in section 1 of the Results, the authors present first the community, then move to the wider framework of how the legislative landscape impacts, then back down to the workplace (including the patient level) perspective. Perhaps starting at the widest legal and administrative context, then moving to the community/media/personal circles and then on to the workplace environment would help the reader move between the locations identified in the findings. This approach has been adopted in section 4.1 of the Discussion where it works well.

Thank you for these comments and for your advice in how we can work towards resolving this issue. In the revised manuscript, we have reduced the number of sub-sections in the Results to, we hope, focus more on the core findings. We think this also lends to ease of reading and makes the section much easier to follow. As regards the Figures and Tables, we have revised to ensure that they correspond with the revised thematic structure. This includes the removal of Figure 3.

Role of geographic locations: Throughout the Results there is limited analysis of how the geographical locations of the studies impact on the results presented. Numerous paragraphs include multiple references to the countries from which the studies were conducted, however the locations do not seem to have much of a bearing on the findings presented. E.g., 1: “Section 3.2.3.4.2 Access to training” covers a range of training topics (including to increase the number of abortion providers available, for technical skill development, to provide psychological support, for improved after-care, and to navigate legislation or ethical and moral issues). However, no links between the types of training identified and the locations the paragraph refers to are made. E.g., 2: “3.3.3.1 Difficult emotions in abortion work” flips from geographical regions to individual countries and back again, and it is unclear why regions or individual countries are being identified to support the findings presented.

We agree that geographic locations and the cultural setting of each country has a unique influence on the experiences of those who provide abortion care. However, it was beyond the possibility of this review to conduct a cross-cultural analysis of providers’ experiences for two main reasons: 1) owing to time constraints and 2) given the exclusion of non-English language articles and the small number of studies for many of the countries. Additionally, the aim of the review was to map experiences evident within the published, peer-reviewed literature and to acknowledge that they may exist, not to examine why each of these experiences occurs.

We do, however, acknowledge that we should have been clearer about this, and we have revised many parts of the article in this regard. 

Firstly, we acknowledge at the beginning of the “Methods and materials” section that the intention of the review is to map the known experiences that providers may share, and that the results may lead to systematic reviews which explore these in more detail (L73-80)

In the results section, we acknowledge that a detailed cross-cultural analysis of the studies will not be presented in our review due to time constraints (L208-210)

And in the limitations section, we also explain why a cross-cultural analysis has not been presented (L735-741). We also state that future reviews that would focus more specifically on aspects of providers’ experiences identified in the current review would be better suited to conducting a cross-cultural analysis. Using the example you provided us with, we propose a review looking only at providers’ experiences with training programs. We also state that a review looking at research from a specific country or region, or research on a specific aspect of providers’ experiences would be more suited to explore the cultural differences, whereas this level of detail for each experience is beyond the current review.

Use of precise and accurate language: Throughout the Results, please ensure that findings are precisely and accurately presented. On occasion, the authors selection of specific language conveys different meaning to the findings than may actually have been identified. E.g: “Difficult emotions were also linked to restrictive policies and practices in abortion care, such as a nurse in Australia who felt the need to break the rules to provide optimum care: “I was scared, wasn’t sleeping, hardly eating, started smoking… but what sustained me was that I knew I was doing the right thing and if I walked away from this and did nothing, then that would be a lot worse. I couldn’t do that.”” It is unclear to me how the nurse in Australia has broken any rules.

Thank you for pointing this point. We have revised the results section to ensure that the results are precisely represented. We have removed the example that you provided in an effort to reduce word count and focus on the key findings. 

Use of quotes: I greatly appreciated the use of direct quotes from abortion providers both in text and separate paragraphs throughout the Results section, striking a good balance in the ratio of text to quotes. They provide appropriate emphasis and illustration of findings. However, with the longer separate paragraphs, I would advise the authors to consider using a wider range of supporting quotes than those currently selected. Of the 12 longer quotes included, 10 are from the USA and 2 from Australia. Whilst there is a concentration of research in the USA, it is important to draw in voices from across the abortion provider landscape.

Thank you for raising this oversight on our side! We agree and ha

---

## [Decision Letter · Decision Letter 1]

29 Apr 2024

Providers’ experiences with abortion care: A scoping review

PONE-D-22-24285R1

Dear Dr. Dempsey,

We’re pleased to inform you that your manuscript has been judged scientifically suitable for publication and will be formally accepted for publication once it meets all outstanding technical requirements.

Kind regards,

Sylvester Chidi Chima, M.D., L.L.M, LLD

Academic Editor

PLOS ONE

Reviewers' comments:

Reviewer's Responses to Questions

**Comments to the Author**

1. If the authors have adequately addressed your comments raised in a previous round of review and you feel that this manuscript is now acceptable for publication, you may indicate that here to bypass the “Comments to the Author” section, enter your conflict of interest statement in the “Confidential to Editor” section, and submit your "Accept" recommendation.

Reviewer #3: All comments have been addressed

2. Is the manuscript technically sound, and do the data support the conclusions?

Reviewer #3: Yes

3. Has the statistical analysis been performed appropriately and rigorously? 

Reviewer #3: Yes

4. Have the authors made all data underlying the findings in their manuscript fully available?

Reviewer #3: Yes

5. Is the manuscript presented in an intelligible fashion and written in standard English?

Reviewer #3: Yes

6. Review Comments to the Author

Reviewer #3: As an abortion provider, I enjoyed this review and found it resonated with my experience. The manuscript is well written and the findings are valuable.

7. PLOS authors have the option to publish the peer review history of their article (what does this mean?). If published, this will include your full peer review and any attached files.

Reviewer #3: No

---

## [Editor Report · Acceptance letter]

29 May 2024

PONE-D-22-24285R1 

PLOS ONE

Dear Dr. Dempsey, 

I'm pleased to inform you that your manuscript has been deemed suitable for publication in PLOS ONE. Congratulations! Your manuscript is now being handed over to our production team.

Kind regards, 

on behalf of

Professor Sylvester Chidi Chima 

Academic Editor

PLOS ONE